# Dietary Supplementation with Mono-Lactate Glyceride Enhances Intestinal Function of Weaned Piglets

**DOI:** 10.3390/ani13081303

**Published:** 2023-04-11

**Authors:** Hanbo Li, Yanyan Zhang, Jiaqian Xie, Chao Wang, Dan Yi, Tao Wu, Lei Wang, Di Zhao, Yongqing Hou

**Affiliations:** Engineering Research Center of Feed Protein Resources on Agricultural By-Products, Ministry of Education, Wuhan Polytechnic University, Wuhan 430023, China

**Keywords:** weaned piglets, mono-lactate glyceride, growth performance, diarrhea, intestine function

## Abstract

**Simple Summary:**

The aim of this study is to investigate the effects of mono-lactate glyceride on growth performance and the morphology and function of the intestine in weaned piglets, which provided a theoretical basis for its practical application as a new feed additive. Dietary supplementation with 0.6% mono-lactate glyceride (LG) essentially decreased diarrhea rate and the contents of malondialdehyde (MDA) and hydrogen peroxide (H_2_O_2_) in the ileum and jejunum, increased the expression of intestinal tight junction protein (Occludin) and the activities of superoxide dismutase (SOD) and catalase (CAT) in the ileum and colon, and improved the intestinal morphologic structure. In addition, mono-lactate glyceride supplementation could enhance intestinal mucosal growth, promote intestinal mucosal water and nutrient transport and lipid metabolism, and enhance antiviral and immune function and antioxidant capacity. Overall, these results suggested that dietary supplementation with mono-lactate glyceride could decrease the diarrhea rate.

**Abstract:**

Mono-lactate glyceride (LG) is a short-chain fatty acid ester. It has been shown that short-chain fatty acid esters play an important role in maintaining intestinal structure and function. The aim of this study is to investigate the effects of mono-lactate glyceride on growth performance and intestinal morphology and function in weaned piglets. Sixteen 21-day-old weaned piglets of similar weight were distributed arbitrarily to two treatments: The control group (basal diet) and the LG group (basal diet + 0.6% mono-lactate glyceride). The experiment lasted for 21 days. On day 21 of the trial, piglets were weighed, and blood and intestinal samples were collected for further analysis. Results showed that dietary supplementation with 0.6% mono-lactate glyceride decreased (*p <* 0.05) the diarrhea rate and the contents of malondialdehyde and hydrogen peroxide in the ileum and jejunum and increased (*p <* 0.05) the expression of intestinal tight junction protein (Occludin) and the activities of superoxide dismutase and catalase in the ileum and colon. In addition, mono-lactate glyceride supplementation could enhance intestinal mucosal growth by increasing (*p <* 0.05) the mRNA levels of extracellular regulated protein kinases, promote intestinal mucosal water and nutrient transport and lipid metabolism by increasing (*p <* 0.05) the mRNA levels of b^0,+^ amino acid transporter, aquaporin 3, aquaporin 10, gap junction protein alpha 1, intestinal fatty acid-binding protein, and lipoprotein lipase, enhance antiviral and immune function by increasing (*p <* 0.05) the mRNA levels of nuclear factor kappa-B, interferon-β, mucovirus resistance protein II, 2’-5’-oligoadenylate synthetase-like, interferon-γ, C-C motif chemokine ligand 2, and toll-like receptor 4, and enhance antioxidant capacity by increasing (*p <* 0.05) the mRNA levels of NF-E2-related factor 2 and glutathione S-transferase omega 2 and decreasing (*p <* 0.05) the mRNA level of NADPH oxidase 2. These results suggested that dietary supplementation with mono-lactate glyceride could decrease the diarrhea rate by improving intestinal antioxidant capacity, intestinal mucosal barrier, intestinal immune defense function, and intestinal mucosal water and nutrient transport. Collectively, dietary supplementation with 0.6% mono-lactate glyceride improved the intestinal function of weaned piglets.

## 1. Introduction

Social, environmental, and nutritional changes will reduce piglet feed intake during the critical period of piglet adaptation to the initial diet. [1]. After weaning, the diet is changed from highly digestible liquid milk to more indigestible and complex solid feeds, which can impair the structure and function of the intestine [2]. Thus, the weaning period is one of the most critical developmental stages of the digestive tract of piglets. Symptoms caused by weaning profoundly affect the health of piglets, resulting in reduced growth performance and sometimes death [3]. These effects have resulted in economic losses to pig production and increased public health risks due to the production of pathogenic bacteria-infected pork [4]. Therefore, it is urgent to develop high-quality and safe antibiotic replacement products to improve adverse impacts caused by weaned piglet syndrome and promote piglet intestinal health.

Short-chain fatty acid esters are low molecular weight chemicals formed by the esterification of short-chain fatty acids (SCFAs) and alcohols under acid catalysis, i.e., the substrates are common with chain lengths of less than ten carbon atoms. It is an important substance to provide energy and fat metabolism to the animal body [5]. When the body is in a state of energy demand, short-chain fatty acid esters are gradually hydrolyzed by lipases to free fatty acids and glycerol, released into the blood, and oxidized and utilized by other tissues [6]. In recent years, it has been shown that dietary supplementation with tributyrin can improve the growth performance of weaned piglets and promote the development of immune organs and small intestine of weaned piglets. [7]. Previous research reported that glyceryl butyrate attenuated the inflammatory responses in ETEC-challenged piglets by inhibiting NF-κB/MAPK pathways and modulating gut microbiota, thereby improving the intestinal health of piglets [8].

Studies have shown that SCFA inhibits inflammatory responses by inhibiting immune cell chemotaxis and reducing the release of pro-inflammatory cytokines and reactive oxygen species [9]. As fatty acids, acetic acid, propionic acid, and butyric acid play an important role. Butyric acid can play an anti-inflammatory role by inhibiting the release of IL-12, IL-1β, TNF-α, and NO in monocytes, promoting the expression of IL-10 and reducing the activity of NF-κB [9]. Acetic acid and the propionic acid act as antimicrobial agents by promoting the secretion of defense peptides in the host [10]. Butyrate can increase the secretion of antimicrobial peptides that are intrinsically immune in the host, clearing *Salmonella* enterica before an inflammatory response is triggered [11]. SCFAs are primarily produced not only from food sources but also from microbial fermentation of non-digestible sugar in the colon and cecum [12].

As a short-chain fatty acid ester, mono-lactate glyceride (LG) is water-soluble. When administered parenterally, mono-lactate glyceride is rapidly hydrolyzed to glycerol and free fatty acids in the small intestine [6]. It has been shown that SCFAs can change chemotaxis and phagocytosis of immune cells, induce reactive oxygen species (ROS), and alter cell proliferation and function, which has anti-inflammatory, antitumorigenic, and antimicrobial effects [13]. Therefore, dietary supplementation with SCFAs can effectively replace antibiotics and improve the growth performance of weaned piglets. We hypothesize that LG can attenuate weaning-induced intestinal oxidative stress and inflammatory responses, thereby improving intestinal function in weaned piglets. However, the effects of mono-lactate glyceride on growth performance and intestinal function in weaned piglets remain unclear. In this experiment, the effects of LG supplementation on growth performance and intestinal function in weaned piglets were investigated by detecting the growth indexes, plasma biochemical parameters, intestinal histomorphology, antioxidant parameters, gene expression levels, and protein expression levels in piglets.

## 2. Materials and Methods

### 2.1. Experimental Animals and Design

The animal experiment for this research was approved by the Animal Care and Use Committee of the Hubei Province (WPU201508001). The sixteen healthy crossbred female piglets (Duroc × Landrace × Yorkshire) with similar body weights were weaned at 21 days of age. Each piglet was individually housed in a 1.20 × 1.10 m^2^ steel metabolic cage with eight replicate cages per treatment. After a period of 3 days of adaptation, piglets (average body weight of 6.59 ± 1.19 kg) were assigned randomly into the two treatments: Control group (piglets were fed with the basal diet) and LG group (piglets were fed with the basal diet supplemented with 0.6% mono-lactate glyceride). The basal diet was prepared according to the nutritional needs of NRC (2012) pigs (8~20 kg), and the basal diet nutritional levels were consistent in each treatment group, and its composition and nutritional content are shown in Table 1. On days 7 and 14 of the trial, blood samples were collected from the anterior vena cava of piglets. On day 21 of the trial, each pig was anesthetized with 10% pentobarbital sodium by intramuscular injection at a dose of 80 mg/kg BW and slaughtered 15 min later. Then, the pig‘s abdomen was incised from the sternum to the pubic bone to expose the entire gastrointestinal tract. Blood, intestine, and intestinal contents were collected and stored at −80 °C until assay [14].

### 2.2. Plasma Biochemical Indices

On the 21st day of the experiment, piglets in each group were aseptically bled from the anterior vena cava 1 h after feeding D-xylose, and the blood samples were allowed to stand for 15 min and centrifuged (3000 rpm, 10 min), and the supernatant was taken after the end of centrifugation, that is, plasma was separated, and the blood samples were aliquoted and placed in a −80 °C freezer for testing [15]. Plasma biochemical Indices (ALT, AST, TBIL, TP, ALB, CHOL, BUN, ALP, CK, and GGT) were measured by the Testing and Analysis Center of the Hubei Institute of Pharmaceutical Industry.

### 2.3. Intestinal Morphology and Intestinal Redox Status

To investigate intestinal morphology, paraformaldehyde-fixed jejunum, ileum, and duodenum were dehydrated and embedded in paraffin. Next, 4-µm sections were cut and then stained with hematoxylin and eosin stain. Intestinal morphology was carried out with a light microscope (Leica, Solms, Germany) with the Leica Application Suite image analysis software (Leica, Solms, Germany) [14]. There are 6 villus and crypts that were counted per histological cutting. Intestinal villus height, crypt depth, and villus surface area were measured to calculate the ratio of villus height to crypt depth. The activities of GSH-Px, SOD, and CAT and the contents of MDA and H_2_O_2_ were determined by using commercially available kits (Jiancheng Bioengineering Institute, Nanjing, China).

### 2.4. Expression Levels of Protein

The expression of proteins was analyzed by using Western blotting as described by Hou et al. [16]. The primary antibodies used in this study included Villin, Occludin (rabbit, 1:1000; Cell Signaling Technology, Inc., Danvers, MA, USA), Caspase-3, Bax, MAX1, and β-actin (mouse 1:2000; Sigma-Aldrich Inc., St. Louis, MI, USA). The secondary antibodies used in this study included Anti-mouse (mouse, 1:5000; Zhongshan Golden Bridge Biological Technology Co., Ltd., Beijing, China) and Anti-rabbit (rabbit, 1:3000; Zhongshan Golden Bridge Biological Technology Co., Ltd., Beijing, China). Blots were developed using an enhanced chemiluminescence kit (Amersham Biosciences, Uppsala, Sweden) and then analyzed by an imaging system (Alpha Innotech, New York, NY, USA).

### 2.5. Expression Levels of Genes

The quantification of gene mRNA levels was performed by the real-time PCR method, as described by Yi et al. [17]. The primers used in this study were shown in Table 2. The reference gene was ribosomal protein L 4 (RPL4). SYBR^®^ Premix Ex Taq^TM^ (Takara, Dalian, China) was used for real-time PCR 7500 System Fast Real-Time RT-PCR (Applied Biosystems, Foster City, CA, USA). Results were analyzed by the 2^−ΔΔCt^ method as described [1].

### 2.6. Statistical Analysis

All data were analyzed using Student’s t-test, and data were expressed as mean ± standard deviation. All results were analyzed using SPSS (Version 17.0, SPSS Inc., Chicago, IL, USA). A *p*-value ≤ 0.05 was considered statistically significant.

## 3. Results

### 3.1. Growth Performance

During the experimental period, the average daily gain (ADG), average daily feed intake (ADFI), feed to gain (F/G), and diarrhea rate (DR) were shown in (Table 3). Although there was no significant difference (*p* > 0.05) in ADG and ADFI between control and LG groups, dietary LG supplementation had a tendency to increase ADG. Moreover, compared with the control group, dietary supplementation with 0.6% LG reduced (*p* < 0.05) DR between days 0 to 10, days 10 to 21, and days 0 to 21.

### 3.2. Plasma Biochemical Indices

Plasma biochemical indicators were shown in Appendix A to reflect the metabolic function of piglets. Compared with the control group, alkaline phosphatase (ALP) and blood urea nitrogen (BUN) on day 7 and alanine aminotransferase (ALT) on day 21 in plasma were increased (*p* < 0.05) in the LG group, total protein (TP), total cholesterol (CHOL), and glutamyl transpeptidase (GGT) were decreased (*p* < 0.05) in the LG group on day 7, total bilirubin (TBIL), albumin (ALB), creatine kinase (CK), and glutamyl transaminase (GGT) were decreased (*p* < 0.05) in the LG group on day 14, and creatine kinase (CK) were decreased (*p* < 0.05) in plasma on day 21.

### 3.3. The Effects of Dietary Supplementation with LG on Piglet Intestinal Morphology

Results showed that the villus surface area and the ratio of villus height to crypt depth in the jejunum and the ratio of villus height to crypt depth in the ileum were increased (*p* < 0.05) in the LG group, whereas the crypt depth in the jejunum and ileum and villus surface area in the ileum were decreased (*p* < 0.05), compared with the control group (Table 4).

### 3.4. Intestinal Redox Status

Compared with the control group, LG supplementation increased (*p* < 0.05) the activities of SOD in the ileum and CAT in the colon and decreased (*p* < 0.05) the activity of GSH-Px in the duodenum and jejunum and the contents of MDA in the jejunum and ileum, and H_2_O_2_ in the jejunum (Table 5).

### 3.5. Protein Abundances

Compared with the control group, piglets in the LG group exhibited an increase (*p* < 0.05) in the protein abundances of Occludin and MX1. There were no significant differences in other protein abundances, including Villin, Caspase-3, and Bax (Figure 1).

### 3.6. Gene Expression

Compared with the control group, LG supplementation increased (*p* < 0.05) the mRNA levels of ERK in the jejunum while decreasing (*p* < 0.05) the mRNA levels of Bax in the jejunum in the LG group (Table 6).

Compared with the control group, piglets in the LG group exhibited obvious increases (*p* < 0.05) in the mRNA levels of b^0,+^AT in the colon, and AQP3 in the ileum and colon, GJA1 in the jejunum, and AQP10 in the ileum (Table 7).

Compared with the control group, LG supplementation increased (*p* < 0.05) the mRNA levels of NF-κB, IFN-β, MX1, MX2, TLR4, and OASL in the jejunum and TFF3, NF-κB, IFN-α, IFN-β, MX2, and OASL in the ileum (Table 8).

Compared with the control group, piglets in the LG group exhibited significant increases (*p* < 0.05) in the mRNA levels of IL-1β and IFN-γ in the jejunum and IL-1β, IL-4, CCL-2, and IFN-γ in the ileum, while also exhibiting a decrease in (*p* < 0.05) the mRNA levels of CXCL-9 in the ileum, and of IFN-γ in the colon (Appendix A).

Compared with the control group, LG supplementation increased (*p* < 0.05) the mRNA levels of LPL, Nrf-2, and GSTO2 in the jejunum and LPL and Nrf-2 in the ileum and INSR in the colon, while also decreasing (*p* < 0.05) the mRNA levels of I-FABP in the jejunum and I-FABP and PCK1 in the ileum and I-FABP and NOX2 in the colon (Table 9).

## 4. Discussion

Early weaning causes stress and diarrhea in piglets, which is one of the most challenging problems in the pig industry [1]. Previous research suggested that dietary supplementation with SCFAs could promote intestinal health by improving intestinal absorption and immunity in piglets [18]. Previous studies suggested that dietary supplementation of calcium butyrate significantly reduced diarrhea rates in piglets [19,20]. In good agreement with these studies, our results demonstrated that dietary supplementation with 0.6% LG effectively reduced the diarrhea rate.

Plasma protein synthesized by the liver can be used as an indicator of protein metabolism function, and an increase in total protein in plasma can indicate enhanced immune function [14,21]. In the present study, although dietary LG supplementation decreased the total protein content in plasma on day 7, there was no significant difference at a later stage, which may be because LG helps to improve piglet immunity and promote body protein synthesis. The level of enzyme activity in plasma is an indicator of tissue damage. Alkaline phosphatase (ALP) is a ubiquitous membrane-bound glycoprotein involved in protein phosphorylation and plays a role in the transport of intestinal epithelial cells [22]. The results showed that LG intervention increased plasma ALP activity and may promote protein synthesis in piglets. In addition, ALT, AST, AST/ALT, and ALP in plasma are indicators of liver function, and liver dysfunction causes an increase in AST/ALT activities, which are gradually decreased when liver function is repaired [23,24,25]. In this study, LG intervention decreased AST/ALT activities at day 21, indicating that LG may help repair liver function. Total bilirubin is an indicator of liver function. It has been shown that TBIL content in the plasma is decreased when liver function is enhanced [25]. In good agreement with these studies, our results showed that there was a significant decrease in TBIL levels in the plasma of piglets in the LG group. GGT is an essential enzyme for protein and amino acid metabolism, and it could reflect the injury of various cells by oxygen free radicals [14]. The elevated plasma GGT levels might be an indicator of oxidative stress and liver damage [26]. We found that LG intervention reduced GGT activity in plasma on days 7 and 14, thereby alleviating the stress experienced by piglets at the early weaning stage. Creatine kinase plays an important role in the process of energy metabolism, and its level is often used as an indicator of cardiac and skeletal muscle disease [27]. A previous study suggested that the CK level was increased in blood when the skeletal muscle was compromised. [28]. In this study, we found that LG intervention decreased CK level in plasma on days 14 and 21, indicating that LG may help repair bone damage caused by stress in piglets. These results showed that supplementation with LG could regulate the metabolism of proteins, improve the immune level of piglets, and relieve stress in piglets, which plays a role in protecting functions of the liver and skeletal muscle in piglets.

Insulin receptor (INSR) is a single-pass transmembrane receptor with tyrosine kinase activity, which is primarily involved in cell growth and metabolic homeostasis. Its main function is to mediate IGF-2 and insulin signaling pathways and then regulate the metabolic activity of the body [29]. Lipoprotein lipase (LPL) is one of the key enzymes in the systemic partitioning and metabolism of lipids. It plays an important role in lipid metabolism, transport, and energy metabolism and affects the growth and development of animals [30]. Intestinal fatty acid binding protein (I-FABP) is a key protein in lipid transport and can transport lipids from the intestinal lumen to enterocytes, bind excess fatty acids, and maintain a stable fatty acid pool in epithelial cells [31]. In this study, we found that LG intervention could regulate lipid metabolism, transport, and fat deposition and activate the insulin signaling pathway. Aquaporins (AQPs) are a family of membrane channel proteins, of which AQP3 and AQP10 are important aquaporins, which can rapidly absorb water in the intestinal cavity into the blood and alter the endocrine environment of the intestinal cavity [14,32]. It has been shown that intestinal absorption of basic amino acids mainly transports and absorbs basic amino acids and cystine into epithelial cells through the b^0,+^ system at the brush border, and b^0,+^AT plays an important role in the b^0,+^ system [33]. GJA1 is a gap junction protein that plays an important role in the exchange of nutrients, ions, and cellular regulators between cells [34]. In this study, dietary supplementation with LG significantly increased the mRNA levels of AQP3, AQP10, GJA1, and b^0,+^AT, suggesting that dietary LG supplementation may improve the transport of water and nutrients in intestinal mucosa, promote the intestinal water metabolism, and effectively relieve diarrhea in piglets.

Indicators of intestinal morphology, such as villus height, surface area, crypt depth, and the ratio of villus height to crypt depth, are commonly used to reflect intestinal morphological development and intestinal morphological integrity. Generally, the decrease in crypt depth and the increases in villus height, and the ratio of villus height to crypt depth reflect improved healthy intestinal development and nutrient absorption [35]. We found that the crypt depth in the jejunum of piglets in the LG group was decreased to a greater extent than the villus height, possibly because LG intervention increased the number of mature cells in the intestinal mucosa and promoted the complete development of the intestinal mucosa, while the growth and proliferation of cells or the cell maturation rate in the intestine had a great relationship with the crypt depth and the increase of the cell maturation rate could reduce the crypt depth. Bax is a representative pro-apoptotic protein in the Bcl-2 family, and Bcl-2 protein can block the apoptosis signal transmission system, thereby inhibiting apoptosis [36]. ERK1/2 is involved in processes such as cell proliferation, growth, and apoptosis [37]. These results showed that LG inhibited intestinal mucosal cell apoptosis by decreasing the relative expression of the Bax gene in the jejunal mucosa. Moreover, the ERK1/2 signaling pathway was activated, and intestinal mucosal growth was promoted by regulating the relative expression of the ERK1/2 gene.

NOX2 is widespread in phagocytes and tissues where it can be activated to induce increased ROS and is one of the major sources of ROS [38]. A previous study found that Nrf2 improves abnormal oxidative stress by increasing the expression of antioxidant-related genes [39]. GSTO2 can affect the expression of corresponding active proteins by changing the transcriptional activity of related genes, thereby regulating the activity of related antioxidant enzymes and reducing the negative effects of oxidative stress [40]. In this study, LG intervention significantly increased the relative expression of Nrf-2, GSTO2 gene in the jejunum, and Nrf-2 gene in the ileum and significantly decreased the relative expression of the NOX2 gene in the colon, suggesting that LG can improve the antioxidant function of the body by regulating the expression of antioxidant related genes. Superoxide dismutase and catalase are antioxidant enzymes, both of which are involved in neutralizing ROS reactions, thereby protecting tissue cells from oxidative damage [41]. MDA is a major product of polyunsaturated fatty acid peroxidation, which can induce toxic stress in cells and is a marker for the assessment of oxidative stress levels in biosomes [42]. H_2_O_2_ is the main product of oxidative stress in the body [43]. We found that dietary supplementation with LG increased the activities of SOD in the ileum and CAT in the colon and decreased the contents of MDA in the jejunum and ileum and H_2_O_2_ in the jejunum. These results are consistent with Yu et al.‘s study that supplementation of 500 mg/kg *B. licheniformis* in the diet of weaned piglets enhanced antioxidant capacity [44]. In summary, dietary supplementation of 0.6% LG enhanced the antioxidant capacity of the intestine by regulating the expression of antioxidant-related genes and increasing the activity of intestinal antioxidant enzymes. Of note, dietary LG supplementation improved the activity of SOD in the ileum but decreased the activity of GSH-Px in the duodenum and jejunum. This may be due to the existence of a dynamic balance mechanism in the body’s antioxidant system. When one mechanism is activated, the other may be inhibited [45].

Intestinal trefoil peptide (ITF, i.e., TFF3) plays an important role in maintaining and repairing mucosa, inhibiting tumors, and regulating cell growth and apoptosis in animals [46]. It has been shown that secretion of TFF3 contributes to improving intestinal mucosal morphology, reduces the generation of inflammatory cells, and repairs and maintains intestinal mucosa [47]. Our results are consistent with that study. IFN-α and IFN-β belong to type I interferons, which can induce cells to produce antiviral enzymes to interfere with viral transcription and translation, thereby achieving the effect of inhibiting viral proliferation [48]. MX protein has a wide range of antiviral effects and GTPase activity, MX1 protein inhibits myxovirus replication, and MX2 protein has a strong inhibitory effect on vesicular stomatitis virus [49,50,51]. Zhou et al. found that porcine Mx1 has activity against classical swine fever virus (CSFV) [52]. TLR4 is an important member of TLR, which plays an important role in innate immunity and inflammation by sensing pathogen-associated molecular patterns [53]. It has been shown that TLR4 signaling in macrophages can activate hundreds of genes that contribute to the protection against bacterial infection [54]. Furthermore, 2‘-5’ oligoadenylates synthesis (OAS) is an antiviral protein induced by interferon, of which OASL belongs to this class of proteins [55]. NF-κB can induce the expression of inhibitors of apoptosis (IAP) and certain members of anti-apoptotic Bcl2 by activating the transcription of genes involved in the inhibition of cell death through intrinsic and extrinsic pathways [56]. Occludin protein is an extremely important protein in the TJ, which promotes tight junctions in the intestinal epithelial cell space [57]. In this study, dietary supplementation with LG increased the mRNA levels of TFF3, NF-κB, IFN-α, IFN-β, MX1, MX2, TLR4, and OASL, as well as the protein expression of Occludin and Mx1. These results supported the notion that mono-lactate glyceride could regulate mucosal protection and repair, regulate apoptosis and anti-virus, and then improve intestinal immunity and intestinal barrier function.

IL-1β is a pro-inflammatory cytokine produced by cells of the innate immune system and is essential in host defense responses [58]. It has been shown that short-chain fatty acids are beneficial in increasing the abundance of IL-1beta and IL-6 in the small intestine without producing intestinal inflammation [59]. IL-4, a Th2 cytokine, is an important regulator of the humoral immune response, which can regulate B cells and other non-immune cells and reflect the cellular and humoral immunity of animals [60,61]. These results showed that LG significantly up-regulated the relative expression of IL-1β in jejunal mucosa and IL-4 and IL-1β genes in ileal mucosa and then regulated Thl/Th2 immune balance, thus, LG played an inflammatory regulatory role. IFN-γ is the only member of the type II interferon family and is mainly produced by activated T cells, and has immunomodulatory functions [62]. Chemokine 9 (C-X-C motif 9, CXCL9), a member of the CXC family of chemokines, has the induction and chemotaxis of T cells and monocytes [63]. The C-C motif chemokine ligand 2 (CCL2) is a crucial mediator of immune cell recruitment during microbial infections and tissue damage [64]. Ferrari et al. found that IFN-γ had a synergistic effect on the secretion of CCL2 [65]. In this study, LG intervention significantly up-regulated the relative expression of CCL2 in ileal mucosa and IFN-γ gene in jejunoileal mucosa, indicating that glyceryl mono-lactate can regulate the immune function of the body.

## 5. Conclusions

Dietary supplementation with 0.6% LG significantly reduced diarrhea by improving intestinal histomorphology, maintaining intestinal integrity, and promoting the intestinal antioxidant capacity and mucosal barrier function, thereby improving intestinal function. Taken together, our results demonstrate the importance of LG in improving gut health in weaned piglets.

## Figures and Tables

**Figure 1 animals-13-01303-f001:**
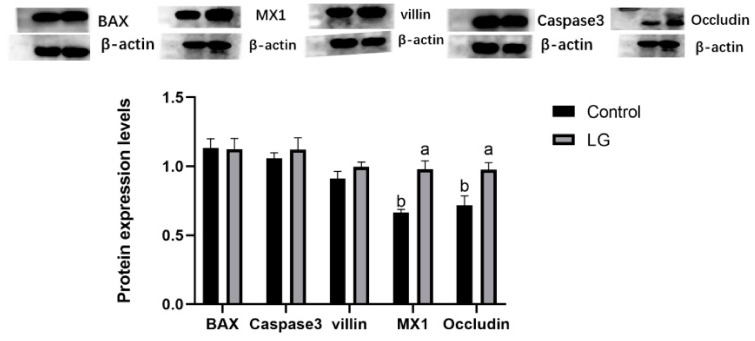
Effect of mono-lactate glyceride (LG) on relative expression of Villin, Occludin, Caspase3, Bax, and MX1 in jejunum of weaned piglets Values are mean and SD, n = 8. ^a,b^ Values within a row with different letters differ (*p* < 0.05). Original western blot figures in Appendix A.

**Table 1 animals-13-01303-t001:** Basal diet composition and nutrient content (air-dried basis) %.

Ingredient	Content	Nutrient Level	Content
Corn	38.40	DE (MJ/kg)	14.27
Soybean meal	16.00	CP	18.54
Flour	12.00	Lys	1.50
Whey powder (low Protein)	8.00	Met	0.42
Soybean protein concentrate	5.00	Met + Cys	0.71
Wheat middling	5.00	Thr	0.93
Fish meal	4.50	Trp	0.23
Glucose	3.00	Ca	0.75
CaHPO_4_	1.33	AP	0.49
Limestone	0.37	TP	0.68
NaCl	0.25	Na	0.31
Plant oil	3.95	CF	6.26
^a^ Premix	1.00	NaCl	0.61
Lys	0.64		
Met	0.13		
Thr	0.21		
Choline	0.12		
Mildew preventive	0.10		

^a^ Premix is provided per kg ration: Fe 100 mg; Cu 150 mg; Mn 40 mg; Zn 100 mg; I 0.5 mg; Se 0.3 mg; VA 1800 IU; VD3 4000 IU; VE 40 IU; VK 34 mg; Thiamine (VB1) 6 mg; Riboflavin (VB2) 12 mg; Pyridoxine (VB6) 6 mg; Cobalamin (VB12) 0.05mg; Biotin 0.2 mg; Folic acid 2 mg; Niacin 50 mg; Calcium pantothenate 5 mg.

**Table 2 animals-13-01303-t002:** Primers for real-time PCR analysis.

Genes	Forward Sequences	Reverse Sequences
Bcl-xl	TGAATCAGAAGCGGAAACCC	GCTCTAGGTGGTCATTCAGGTAAG
ERK	AAGCTCTTGAAGACGCAGCAC	CAGCAGGTTGGAAGGTTTGAG
Bax	TTTCTGACGGCAACTTCAACTG	AGCCACAAAGATGGTCACTGTCT
b^0,+^AT	CGAGTACCCGTACCTGATGGA	TGCGTAGAAGGGCGAAGAA
AQP3	AAGCTGTCCCAAGTAAAGCACAA	GCCCTACTTCCTGTTTCACCAC
GJA1	CAAATCCTTCCCCATCTCTCAC	TCAGTTTCTCTTCCTTTCGCATC
AQP10	TGTCTGCTTTCTGTGCCTCTG	GGATGCCATTGCTCAAGGATAGATAA
TFF3	AGTGTGCCGTCCCTGCCAAG	GCAGCCCCGGTTGTTGCACT
NF-κB	CTCGCACAAGGAGACATGAA	ACTCAGCCGGAAGGCATTAT
IFN-α	ACTCCATCCTGGCTGTGAGGAAAT	ATCTCATGACTTCTGCCCTGACGA
IFN-β	ATGTCAGAAGCTCCTGGGACAGTT	AGGTCATCCATCTGCCCATCAAGT
MX1	AGTGCGGCTGTTTACCAAG	TTCACAAACCCTGGCAACTC
MX2	CGCATTCTTTCACTCGCATC	CCTCAACCCACCAACTCACA
TLR4	GCCTTTCTCTCCTGCCTGAG	AGCTCCATGCATTGGTAACTAATG
OASL	GGCACCCCTGTTTTCCTCT	AGCACCGCTTTTGGATGG
IL-1β	CAACGTGCAGTCTATGGAGT	GAGGTGCTGATGTACCAGTTG
IL-4	TACCAGCAACTTCGTCCAC	ATCGTCTTTAGCCTTTCCAA
IFN-γ	TCTGGGAAACTGAATGACTTCG	GACTTCTCTTCCGCTTTCTTAGGTT
CCL-2	CATAAGCCACCTGGACAAGAAAA	GGGTATTTAGGGCAAGTTAGAAGGA
CXCL-9	CTTGCTTTTGGGTATCATCTTCCT	TCATCCTTTGGCTGGTGTTG
LPL	AGCCTGAGTTGGACCCATGT	CTCTGTTTTCCCTTCCTCTCTCC
I-FABP	AGATAGACCGCAATGAGA	TCCTTCTTGTGTAATTATCATCAGT
INSR	GGGGCTAAAGAGGAACTATGAGG	AGAGGAAAGCGAAGACAGGAAA
PCK1	CGGGATTTCGTGGAGA	CCTCTTGATGACACCCTCT
Nrf2	GAAGTGATCCCCTGATGTTGC	ATGCCTTCTCTTTCCCCTATTTCT
NOX2	TGTATCTGTGTGAGAGGCTGGTG	CGGGACGCTTGACGAAA
GSTO2	GCCTTGAGATGTGGGAGAGAA	AAGATGGTGTTCTGATAGCCAAGA
RPL4	GAGAAACCGTCGCCGAAT	GCCCACCAGGAGCAAGTT

**Table 3 animals-13-01303-t003:** The effects of dietary supplementation with mono-lactate glyceride (LG) on the growth performance and diarrhea rate of piglets.

Items	Control	LG	*p*-Value
0–10 days			
ADG (g)	239.40 ± 84.79	262.60 ± 46.96	0.459
ADFI (g)	295.35 ± 70.08	296.18 ± 30.61	0.981
F/G	1.28 ± 0.21	1.15 ± 0.23	0.390
DR (%)	27.00 ^a^	13.00 ^b^	0.013
10–21 days			
ADG (g)	403.27 ± 119.00	425.82 ± 82.12	0.628
ADFI (g)	592.00 ± 122.76	634.09 ± 46.56	0.494
F/G	1.48 ± 0.06	1.51 ± 0.21	0.770
DR (%)	11.80 ^a^	2.70 ^b^	0.009
0–21 days			
ADG (g)	325.24 ± 97.15	348.10 ± 43.53	0.506
ADFI (g)	450.74 ± 96.12	473.18 ± 29.66	0.631
F/G	1.40 ± 0.10	1.37 ± 0.16	0.697
DR (%)	19.00 ^a^	7.60 ^b^	0.001

Values are mean and SD, n = 8. ^a,b^ Values within a row with different letters differ (*p* < 0.05).

**Table 4 animals-13-01303-t004:** The effects of dietary supplementation with mono-lactate glyceride (LG) on piglet intestinal morphology.

Item	Jejunum			Ileum		
	Control	LG	*p*-Value	Control	LG	*p*-Value
Villus height (µm)	304.4 ± 26.8	335.0 ± 40.1	0.075	237.6 ± 26.8	221.8 ± 32.8	0.281
Crypt depth (µm)	223.2 ± 19.0 ^a^	180.9 ± 12.9 ^b^	<0.001	168.9 ± 17.9 ^a^	140.0 ± 17.0 ^b^	0.003
Ratio of villus height/crypt depth	1.37 ± 0.08 ^b^	1.85 ± 0.20 ^a^	<0.001	1.41 ± 0.09 ^b^	1.58 ± 0.06 ^a^	<0.001
Villus surface area (µm^2^)	27,127 ± 2917	30,727 ± 4186	0.050	26,287 ± 750 ^a^	23,224 ± 3256 ^b^	0.014

Values are mean and SD, n = 8. ^a,b^ Values within a row with different letters differ (*p* < 0.05).

**Table 5 animals-13-01303-t005:** The effects of dietary supplementation with mono-lactate glyceride (LG) on intestinal redox status of piglets.

Items	Control	LG	*p*-Value	Control	LG	*p*-Value
	Duodenum			Ileum		
SOD (U/mg)	42.25 ± 3.76	42.11 ± 3.09	0.939	84.16 ± 6.22 ^b^	110.03 ± 23.44 ^a^	0.009
GSH-Px (U/mg)	45.82 ± 8.31 ^a^	32.56 ± 6.40 ^b^	0.003	73.60 ± 16.46	87.31 ± 14.84	0.161
CAT (U/mg)	17.92 ± 4.41	16.11 ± 2.39	0.325	9.54 ± 3.08	8.84 ± 2.80	0.641
H_2_O_2_ (nmol/mg)	5.06 ± 1.11	5.16 ± 1.36	0.869	8.34 ± 1.80	9.81 ± 2.45	0.191
MDA (nmol/mg)	3.99 ± 1.18	5.06 ± 1.50	0.136	6.57 ± 1.78 ^a^	4.66 ± 1.12 ^b^	0.022
	Jejunum			Colon		
SOD (U/mg)	85.39 ± 11.40	93.66 ± 14.66	0.228	100.54 ± 25.07	112.40 ± 17.82	0.294
GSH-Px (U/mg)	14.01 ± 4.74 ^a^	8.40 ± 1.99 ^b^	0.014	37.31 ± 7.15	30.44 ± 10.52	0.215
CAT (U/mg)	7.64 ± 1.22	7.98 ± 1.05	0.654	6.56 ± 1.15 ^b^	9.21 ± 1.62 ^a^	0.018
H_2_O_2_ (nmol/mg)	28.16 ± 7.23 ^a^	20.46 ± 5.14 ^b^	0.028	25.25 ± 6.43	27.59 ± 6.81	0.491
MDA (nmol/mg)	10.55 ± 3.45 ^a^	4.73 ± 1.32 ^b^	0.001	-	-	-

Values are mean and SD, n = 8. ^a,b^ Values within a row with different letters differ (*p* < 0.05).

**Table 6 animals-13-01303-t006:** The effects of mono-lactate glyceride (LG) supplementation on the mRNA levels of genes related to proliferation and growth of jejunal mucosa in piglets.

Items	Control	LG	*p*-Value
Bc1-x1	1.00 ± 0.09	1.10 ± 0.17	0.172
ERK	1.00 ± 0.10 ^b^	1.16 ± 0.07 ^a^	0.003
Bax	1.00 ± 0.12 ^a^	0.78 ± 0.16 ^b^	0.010

Values are mean and SD, n = 8. ^a,b^ Values within a row with different letters differ *p* < 0.05).

**Table 7 animals-13-01303-t007:** The effects of mono-lactate glyceride (LG) supplementation on mRNA levels of genes related to transport channel-related in piglets.

Items	Jejunum			Ileum			Colon		
	Control	LG	*p*-Value	Control	LG	*p*-Value	Control	LG	*p*-Value
b^0,+^AT	-	-	-	-	-	-	1.00 ± 0.22 ^b^	1.62 ± 0.40 ^a^	0.002
AQP3	-	-	-	1.00 ± 0.22 ^b^	2.32 ± 0.39 ^a^	<0.001	1.00 ± 0.23 ^b^	1.67 ± 0.37 ^a^	0.001
GJA1	1.00 ± 0.20 ^b^	1.21 ± 0.18 ^a^	0.040	1.00 ± 0.18	0.84 ± 0.14	0.071	-	-	-
AQP10	-	-	-	1.00 ± 0.22 ^b^	2.99 ± 0.76 ^a^	<0.001	-	-	-

Values are mean and SD, n = 8. ^a,b^ Values within a row with different letters differ (*p* < 0.05).

**Table 8 animals-13-01303-t008:** The effects of mono-lactate glyceride (LG) supplementation on mRNA levels of genes related to intestinal mucosal immunity in piglets.

Items	Jejunum			Ileum			Colon		
	Control	LG	*p*-Value	Control	LG	*p*-Value	Control	LG	*p*-Value
TFF3	1.00 ± 0.20	0.86 ± 0.12	0.109	1.00 ± 0.15 ^b^	1.26 ± 0.27 ^a^	0.029	1.00 ± 0.13	1.06 ± 0.21	0.536
NF-κB	1.00 ± 0.10 ^b^	1.13 ± 0.07 ^a^	0.009	1.00 ± 0.10 ^b^	1.22 ± 0.07 ^a^	<0.001	1.00 ± 0.10	1.07 ± 0.10	0.211
IFN-α	1.00 ± 0.20	0.88 ± 0.10	0.143	1.00 ± 0.16 ^b^	1.55 ± 0.36 ^a^	0.002	1.00 ± 0.25	0.83 ± 0.15	0.126
IFN-β	1.00 ± 0.26 ^b^	1.67 ± 0.38 ^a^	0.001	1.00 ± 0.24 ^b^	1.69 ± 0.42 ^a^	0.001	1.00 ± 0.14	1.02 ± 0.17	0.772
MX1	1.00 ± 0.24 ^b^	1.82 ± 0.47 ^a^	0.001	1.00 ± 0.14	1.00 ± 0.21	0.989	1.00 ± 0.17	1.26 ± 0.32	0.057
MX2	1.00 ± 0.27 ^b^	2.39 ± 0.54 ^a^	<0.001	1.00 ± 0.14 ^b^	2.09 ± 0.47 ^a^	<0.001	1.00 ± 0.22	0.89 ± 0.19	0.291
TLR4	1.00 ± 0.14 ^b^	1.23 ± 0.24 ^a^	0.036	1.00 ± 0.05	0.99 ± 0.03	0.796	1.00 ± 0.14	1.04 ± 0.23	0.672
OASL	1.00 ± 0.18 ^b^	2.72 ± 0.59 ^a^	<0.001	1.00 ± 0.25 ^b^	1.90 ± 0.39 ^a^	<0.001	1.00 ± 0.26	0.94 ± 0.20	0.631

Values are mean and SD, n = 8. ^a,b^ Values within a row with different letters differ (*p* < 0.05).

**Table 9 animals-13-01303-t009:** The effects of mono-lactate glyceride (LG) supplementation on the mRNA levels of genes related to energy metabolism in intestinal mucosa and intestinal mucosal oxidation in piglets.

Items	Jejunum			Ileum			Colon		
	Control	LG	*p*-Value	Control	LG	*p*-Value	Control	LG	*p*-Value
LPL	1.00 ± 0.23 ^b^	2.22 ± 0.30 ^a^	<0.001	1.00 ± 0.20 ^b^	1.38 ± 0.33 ^a^	0.014	1.00 ± 0.20	1.05 ± 0.19	0.600
I-FABP	1.00 ± 0.22 ^a^	0.71 ± 0.13 ^b^	0.007	1.00 ± 0.27 ^a^	0.76 ± 0.15 ^b^	0.040	1.00 ± 0.19 ^a^	0.66 ± 0.15 ^b^	0.001
INSR	1.00 ± 0.19	0.87 ± 0.12	0.131	1.00 ± 0.24	1.17 ± 0.26	0.201	1.00 ± 0.20 ^b^	1.29 ± 0.29 ^a^	0.037
PCK1	1.00 ± 0.20	0.86 ± 0.22	0.209	1.00 ± 0.28 ^a^	0.73 ± 0.20 ^b^	0.044	1.00 ± 0.20	1.04 ± 0.19	0.719
NOX2	1.00 ± 0.15	0.97 ± 0.21	0.762	1.00 ± 0.17	1.09 ± 0.26	0.428	1.00 ± 0.19 ^a^	0.77 ± 0.15 ^b^	0.017
Nrf-2	1.00 ± 0.20 ^b^	1.41 ± 0.33 ^a^	0.009	1.00 ± 0.20 ^b^	1.29 ± 0.27 ^a^	0.028	1.00 ± 0.22	0.92 ± 0.18	0.464
GSTO2	1.00 ± 0.13 ^b^	1.34 ± 0.21 ^a^	0.002	1.00 ± 0.09	1.17 ± 0.23	0.074	1.00 ± 0.22	1.05 ± 0.28	0.691

Values are mean and SD, n = 8. ^a,b^ Values within a row with different letters differ (*p* < 0.05).

## Data Availability

The datasets used and/or analyzed during the current study are available from the corresponding authors on reasonable request.

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
