# Peer review of "Dietary Supplementation with Mono-Lactate Glyceride Enhances Intestinal Function of Weaned Piglets"

_animals, 2023, doi:10.3390/ani13081303_

Round 1

Reviewer 1 Report (Previous Reviewer 1)

Thank you for your thoughtful reply.

  • Good research work!

Author Response

Thank you very much!

Reviewer 2 Report (Previous Reviewer 2)

The requested adjustments were done and the paper is appropriate to be published.

Author Response

Thank you very much!

This manuscript is a resubmission of an earlier submission. The following is a list of the peer review reports and author responses from that submission.

Round 1

Reviewer 1 Report

1. The “2.2 Plasma Biochemical Indices section recommends that relevant indicators be listed and references supplemented.
2. 2.3 section. It is suggested to supplement relevant references.
3. It is suggested to supplement the representational morphological structure diagram in Section 3.3.

  •  
  •  
  •  

Author Response

Response to Reviewer 1:

Thank you very much for your valuable comments on our manuscript. We have carefully examined and revised the manuscript according to your comments in the revised manuscript.

Reviewer 1

  1. The “2.2 Plasma Biochemical Indices” section recommends that relevant indicators be listed and references supplemented.

Response:Thanks for the reviewer’s valuable advice. The plasma Biochemical Indices and related literature have been added to the revised manuscript (Lines 133 to 138).

  1. 2.3 section. It is suggested to supplement relevant references.

Response:Thanks for your careful work. The relevant references have been added in the revised manuscript (Line 145).

  1. It is suggested to supplement the representational morphological structure diagram in Section 3.3.

Response:Thanks for the reviewer’s valuable advice. Our laboratory generally accounts for morphological changes by measuring indicators such as villi, as reported in these literatures (10.3390/ijms19072084 [1], 10.1007/s00726-011-1191-9 [2], 10.1007/s00726-018-2586-7 [3] ), so pictures of intestinal morphology were not used in this study.

[1] Wu T, Zhang Y, Lv Y, Li P, Yi D, Wang L, Zhao D, Chen H, Gong J, Hou Y. Beneficial Impact and Molecular Mechanism of Bacillus coagulans on Piglets' Intestine. Int J Mol Sci. 2018 Jul 18;19(7):2084.

[2] Hou Y, Wang L, Zhang W, Yang Z, Ding B, Zhu H, Liu Y, Qiu Y, Yin Y, Wu G. Protective effects of N-acetylcysteine on intestinal functions of piglets challenged with lipopolysaccharide. Amino Acids. 2012 Sep;43(3):1233-42.

[3] Yi D, Li B, Hou Y, Wang L, Zhao D, Chen H, Wu T, Zhou Y, Ding B, Wu G. Dietary supplementation with an amino acid blend enhances intestinal function in piglets. Amino Acids. 2018 Aug;50(8):1089-1100.

Reviewer 2 Report

SIMPLE SUMMARY

Lines 15 to 21: the text is very similar. You say the same thing two times. Try to change it.

INTRODUCTION

-lines 75 and 76: the examples should be with pigs, not cows. There are other papers related with this specie.

-lines 82 and 84: the authors should be highlight butyrate, acetic and propionic acid as Fat acid

MATERIALS AND METHODS

- line 106: were the pigs of the same sex? Explain it please.

- line 112: Explain how was done the slaughter of the piglets

- Insert the Table with the rations composition with their nutritional levels

- line 116: What are the Biochemical Indices? Present them please

- line 124: How much villus and crypts were counted per histological cutting

RESULTS

- lines 152 to 154: it is important to show the p value

- all Tables: would be intersting to present the p value for all parameters

- In all Tables: insert in the final of the titles (means and standart deviation)

- line 173 and 174. Remove this phrase

- Table 4: the unit of villus surface is wrong (cm2 is impossible)

- line 182 and 183: remove the frase.

DISCUSSION

- Overall, the authors should discuss how the product act and for this reason explain the results

- The way the authors proposed to discuss the results is tiring and repetitive. Many explanations of what the evaluated parameters represent are known and could be avoided, making the text lighter. For this reason some references are old.

- I suggest that searches be made for articles that have done something similar to also validate the effects of the tested product or similar products in swine.

Author Response

Response to Reviewer 2:

Thank you very much for your valuable comments on our manuscript. We have carefully examined and revised the manuscript according to your comments in the revised manuscript.

Reviewer 2

SIMPLE SUMMARY

  1. Lines 15 to 21: the text is very similar. You say the same thing two times. Try to change it.

Response:Thanks for the reviewer’s valuable advice. Repeated parts have been removed from the original manuscript (line 19 to 21).

INTRODUCTION

  1. lines 75 and 76: the examples should be with pigs, not cows. There are other papers related with this specie.

Response:Thanks for your careful work. Replaced references related to swine and added to revised manuscript (line 76 to 80).

  1. lines 82 and 84: the authors should be highlight butyrate, acetic and propionic acid as Fat acid.

Response:Thanks for your careful work. We highlight butyrate, acetic acid and propionic acid as fatty acids in the revised manuscript (line 86 to 87).

MATERIALS AND METHODS

  1. line 106: were the pigs of the same sex? Explain it please.

Response:Thanks for your careful work. According to your comments, the sex of the pig has already been described in more detail in the revised manuscript (line 114).

  1. line 112: Explain how was done the slaughter of the piglets.

Response:Thanks for your careful work. According to your comments, the piglet slaughter process has already been described in more detail in the revised manuscript.

  1. Insert the Table with the rations composition with their nutritional levels.

Response:Thanks for your careful work. We inserted "rations components and their nutrient levels" into the revised manuscript in tabular form (line 130 to 131).

  1. line 116: What are the Biochemical Indices? Present them please.

Response:Thanks for your careful work. We listed all parameters of plasma biochemistry and added them to the revised manuscript (line 137 to 138).

  1. line 124: How much villus and crypts were counted per histological cutting?

Response:Thanks for your careful work. According to your comments, villus and crypts has already been described in more detail in the revised manuscript (line 145 to 146).

RESULTS

  1. lines 152 to 154: it is important to show the p value.

Response:Thanks for your careful work. We inserted P-values in each table and inserted them into the revised manuscript.

  1. all Tables: would be intersting to present the p value for all parameters.

Response:Thanks for your careful work. We inserted P-values in each table and inserted them into the revised manuscript.

  1. In all Tables: insert in the final of the titles (means and standart deviation)

Response:Thanks for your careful work. We inserted "mean and standard deviation" at the end of the titles of all tables and inserted them in the revised manuscript.

  1. line 173 and 174. Remove this phrase.

Response:Thanks for your careful work. According to your comments, we have deleted this phrase (line 196 to 197).

  1. line 182 and 183: remove the phrase.

Response:Thanks for your careful work. According to your comments, we have deleted this phrase (line 206 to 207).

  1. Table 4: the unit of villus surface is wrong (cm2 is impossible)

Response:Thanks for your careful work. According to your comments, we have corrected the wrong unit symbol (line 203 to 204).

DISCUSSION

  1. Overall, the authors should discuss how the product act and for this reason explain the results.

Response:Thanks for your careful work. Following your suggestion, we rewrite the discussion section in the revised manuscript (line 262 to 462).

  1. The way the authors proposed to discuss the results is tiring and repetitive. Many explanations of what the evaluated parameters represent are known and could be avoided, making the text lighter. For this reason, some references are old.

Response:Thanks for your careful work. Following your suggestion, we rewrite the discussion section in the revised manuscript (line 262 to 462).

  1. I suggest that searches be made for articles that have done something similar to also validate the effects of the tested product or similar products in swine.

Response:Thanks for your careful work. Following your suggestion, we rewrite the discussion section in the revised manuscript (line 262 to 462).

Reviewer 3 Report

Review to:

Dietary supplementation with mono-lactate glyceride enhances intestinal function of weaned piglets”

Dear authors,

Thank you for the manuscript dealing with an interesting theme. However, besides the English language that should be revised by a native speaker, the discussion needs substantial revision. Only after total revision of the discussion the manuscript can become acceptable for publication.

Introduction

p.2, l.58-59: In my opinion the reference does not fit to the statement. Please correct with a reference clearly dealing with piglet feeding and pig production itself.

p.3, 87-88: I do not understand the sentence: ”SCFAs are mainly end-products of non-peptic sugars that are fermentative enzymes by microorganisms in the colon and cecum.” Why do you think non-peptic sugars would be fermentative enzymes? Please explain this and change the sentence to become better understandable.

p.3, l. 96-98: should be changed as follows: “However, the effects of mono-lactate glyceride on growth performance and intestinal function in weaned piglets remains unclear.”

A hypothesis is missing in the manuscript. Please include a clear hypothesis.

Material and Methods

p.4, l. 139: should be: “were shown”

Results

Table 2, 3, 4 and 5: Please add the number of pigs per group into the headline. The headlines of tables have to be self explaining without reading the whole manuscript.

p.7, l. 192: should be “Compared”

Figure 1: Please ad a unit to the scale. Additionally please give a more detailed description: E.g. … Protein expression…. in piglets in the control group compared to LG group, n=??

In my opinion some of the tables like table 3 and 9 should be deleted from the manuscript and added to supplemental material.

Table 6 , 7, 8, 9 and 10: Again, please correct the headline to a more detailed one and add the number of animals used

Discussion

p.9, l.239: Please delete “function” as absorption function is not a known term.

p.10, l. 245: Please delete “function” as metabolism function is also not a known term.

p.10, ll. 247-248: “plays a role in the trafficking of intestinal epithelial cells.” Please correct this part of the sentence, as I do not understand it. “Trafficking” does not seem to be the correct term.

p.10, ll.248-249: Do you really mean liver injury? Or do you mean liver function or liver disease? In my opinion, liver injury would be mentioned here without any connection to the study.

p.10, ll.250-252: Please delete the following sentence and reference 26: “Cholesterol in the blood…. can affect the health of animals.” This sentence has nothing to do with the weaned piglets and should not be used in the given discussion.

Please revise the whole discussion as beginning with the chapter on the blood parameters from p.10, ll. 234-262 it is not really a discussion but more a literature review. A lot of statements are given that have nothing to do with healthy or ill piglets. Please change the whole discussion in direction to answer your hypothesis and delete all unnecessary parts dealing with physiological or pathophysiological parameters without connection to diarrhea in weaned piglets.

The same is true for the next part: p.10ll263-281: This part is also not discussion but only a literature review on different parameters. You do not discuss anything in this part. For this reason it has to be deleted from the discussion.

Also from ll. 282-308 this is not discussion but only literature review. Please delete this part.  The same is true for the part p.11, ll-.320-342 and p.12, ll.348-360. Please revise the discussion totally to delete the literature review and to discuss your own results with results from literature. It is necessary to describe the connection between own results and literature results.

References

The references 12 and 13 have to be exchanged due to their first use in the manuscript. Reference 19 should be 14 due to the first use in the text. The following references have to be exchanged accordingly.

After reference 26 was deleted the following references have to be adjusted. 

Author Response

Response to Reviewer 3:

Thank you very much for your valuable comments on our manuscript. We have carefully examined and revised the manuscript according to your comments in the revised manuscript.

Reviewer 3

Introduction

  1. p.2, l.58-59: In my opinion the reference does not fit to the statement. Please correct with a reference clearly dealing with piglet feeding and pig production itself.

Response:Thanks for the reviewer’s valuable advice. The references used here are not problematic, and we adapted sentence meaning in the revised manuscript to align it with the references (line 58 to 60).

  1. p.3, 87-88: I do not understand the sentence: “SCFAs are mainly end-products of non-peptic sugars that are fermentative enzymes by microorganisms in the colon and cecum.” Why do you think non-peptic sugars would be fermentative enzymes? Please explain this and change the sentence to become better understandable.

Response:Thanks for the reviewer’s valuable advice. This sentence is problematic in writing, and we have changed it to "SCFAs are not produced only from food sources but also from microbial fermentation of non-digestible sugar in the colon and cecum" in the revised manuscript (line 92 to 95).

  1. p.3, l. 96-98: should be changed as follows: “However, the effects of mono-lactate glyceride on growth performance and intestinal function in weaned piglets remains unclear.”

Response:Thanks for the reviewer’s valuable advice. We have revised your proposal in the revised manuscript (line 104 to 106).

  1. A hypothesis is missing in the manuscript. Please include a clear hypothesis.

Response:Thanks for the reviewer’s valuable advice. We added a hypothesis "We hypothesize that LG can attenuate weaning-induced intestinal oxidative stress and inflammatory responses, thereby improving intestinal function in weaned piglets." (line 102 to 104)

Material and Methods

  1. p.4, l. 139: should be: “were shown”

Response:Thanks for your careful work. We modified the error here in the revised manuscript (line 162).

Results

  1. Table 2, 3, 4 and 5: Please add the number of pigs per group into the headline. The headlines of tables have to be self explaining without reading the whole manuscript.

Response:Thanks for your careful work. We describe each table in more detail in the revised manuscript and add the number of pigs per group to the title.

  1. p.7, l. 192: should be “Compared”

Response:Thanks for your careful work. According to your comments, we modified the error here in the revised manuscript (line 217).

  1. Figure 1: Please ad a unit to the scale. Additionally, please give a more detailed description: E.g. … Protein expression…. in piglets in the control group compared to LG group, n=??

Response:Thanks for your careful work. According to your comments, we rerevised the narrative of Figure 1 in the revised manuscript (line 221 to 223).

  1. In my opinion some of the tables like table 3 and 9 should be deleted from the manuscript and added to supplemental material.

Response:Thanks for your careful work. Tables 3 and 9 have been removed from the revised version and are presented as attachments.

  1. Table 6, 7, 8, 9 and 10: Again, please correct the headline to a more detailed one and add the number of animals used.

Response:Thanks for your careful work. We describe each table in more detail in the revised manuscript and add the number of pigs per group to the title.

DISCUSSION

  1. p.9, l.239: Please delete “function” as absorption function is not a known term.

Response:Thanks for your careful work. Following your suggestion, we removed the "function" from this sentence in the revised manuscript (line 265).

  1. p.10, l. 245: Please delete “function” as metabolism function is also not a known term.

Response:Thanks for your careful work. Following your suggestion, we removed the "function" from this sentence in the revised manuscript (line 265).

  1. p.10, ll. 247-248: “plays a role in the trafficking of intestinal epithelial cells.” Please correct this part of the sentence, as I do not understand it. “Trafficking” does not seem to be the correct term.

Response:Thanks for your careful work. We changed trafficking to transport in this sentence in the revised manuscript (line 279).

  1. p.10, ll.248-249: Do you really mean liver injury? Or do you mean liver function or liver disease? In my opinion, liver injury would be mentioned here without any connection to the study.

Response:Thanks for your careful work. We changed "liver injury" to "live function" in this sentence in the revised manuscript (line 282).

  1. p.10, ll.250-252: Please delete the following sentence and reference 26: “Cholesterol in the blood…. can affect the health of animals.” This sentence has nothing to do with the weaned piglets and should not be used in the given discussion.

Response:Thanks for your careful work. Following your suggestion, we have deleted this sentence and deleted the literature26 in the revised manuscript (line 289 to 291).

  1. Please revise the whole discussion as beginning with the chapter on the blood parameters from p.10, ll. 234-262 it is not really a discussion but more a literature review. A lot of statements are given that have nothing to do with healthy or ill piglets. Please change the whole discussion in direction to answer your hypothesis and delete all unnecessary parts dealing with physiological or pathophysiological parameters without connection to diarrhea in weaned piglets.

Response:Thanks for your careful work. Following your suggestion, we rewrite the discussion section in the revised manuscript (line 262 to 462).

  1. The same is true for the next part: p.10ll263-281: This part is also not discussion but only a literature review on different parameters. You do not discuss anything in this part. For this reason it has to be deleted from the discussion.

Response:Thanks for your careful work. Following your suggestion, we rewrite the discussion section in the revised manuscript (line 262 to 462).

  1. Also from ll. 282-308 this is not discussion but only literature review. Please delete this part. The same is true for the part p.11, ll-.320-342 and p.12, ll.348-360. Please revise the discussion totally to delete the literature review and to discuss your own results with results from literature. It is necessary to describe the connection between own results and literature results.

Response:Thanks for your careful work. Following your suggestion, we rewrite the discussion section in the revised manuscript (line 262 to 462).

References

  1. The references 12 and 13 have to be exchanged due to their first use in the manuscript. Reference 19 should be 14 due to the first use in the text. The following references have to be exchanged accordingly.

Response:Thanks for your careful work. Following your suggestion, we re-numbered the articles sequentially in the revised manuscript (line 95 and 101).

  1. After reference 26 was deleted the following references have to be adjusted. 

Response:Thanks for your careful work. Following your suggestion, we re-numbered the articles sequentially in the revised manuscript (line 485 to 676).

Round 2

Reviewer 2 Report

The all recommendations were implanted, and the manuscript was improved, deserving publication.

Reviewer 3 Report

Dear authors,

thank you for the careful revision of the manuscript.